# Tracking Boats on Amazon Rivers—A Case Study with the LoRa/LoRaWAN

**DOI:** 10.3390/s25020496

**Published:** 2025-01-16

**Authors:** Marlos Rodrigues, Alyson dos Santos, Hillermann Lima, Warley Nogueira, Vicente de Lucena Junior

**Affiliations:** 1Federal Institute of Education, Science, and Technology of Amazonas, Pólo de Inovação Manaus, Manaus 69075-351, AM, Brazil; alyson.santos@ifam.edu.br (A.d.S.); hillermann@ifam.edu.br (H.L.); 2Hana Electronics, Manaus 69075-010, AM, Brazil; warley.nogueira@hanaelectronics.com.br; 3Electronic and Information Technology Research and Development Center (CETELI), Federal University of Amazonas, Manaus 69067-005, AM, Brazil; vicente@ufam.edu.br

**Keywords:** Amazon rainforest, interconnected boats, rural networks, LoRa

## Abstract

The Amazon region has the largest hydrographic basin in the world. The rivers act as roads, and boats serve as vehicles for transporting passengers and cargo to large urban centers, municipalities, riverside communities, villages, and settlements. The Amazon River transportation system faces critical gaps due to the lack of land infrastructure in certain areas, which makes rivers essential for commerce and access to isolated communities. This work proposes the development of a new hardware platform consisting of a system-in-package (SiP)—iMCP HTLRBL32L and GPS, which enables data transfer over a network with long-range LoRa technology. In addition, we developed a new communication protocol between the end devices called the LoRa Protocol Proprietary (LPP). This protocol allows parameterizable commands (location table, modulation, routing, source and destination DevEUI, and port) to be sent between end devices, providing flexibility in the configuration and management of Internet of Things networks. The results of the practical experiments with the new hardware platform in the communication scenario between the end device and the gateway show that the maximum range was 16.928 km. In the communication scenario between the end devices, the maximum range was 12.447 km. It offers a stable platform for exchanging control information, which is fundamental to the safety of river transport in the Amazon.

## 1. Introduction

Nations constantly engage in international trade driven by globalization [1,2,3]. Thus, the continuous increase in trade between nations and the need to connect supply chains make the river environment a highly significant transportation chain [4,5]. It has substantial economic and social repercussions in remote regions and urban complexes. Specifically, the Brazilian Amazon has dense forests [6,7], numerous rivers [8,9,10,11], and lakes, which pose a barrier to the construction of highways and railways.

Instead of land and asphalt, we have the waters of more than 7000 rivers, with over 25,000 km of navigable waterways [12]. Instead of trucks, we have cargo ships and barges [13,14,15]. Regional boats play the role of buses, express boats play the role of cars, and boats with outboard motors (also known as “catraia”) play the role of motorcycles. Thus, the rivers are the roads of the Amazon [15,16,17]. As a result, access to various cities, communities, villages, and settlements in the Amazon is predominantly by rivers [15,18]. Due to its geographical, technological, and socioeconomic characteristics, the Amazon region faces significant obstacles in implementing existing communication technologies. Innovative solutions and optimized network designs are essential to overcoming these barriers and enhancing connectivity in this remote area.

Considering that boats travel very frequently to the most diverse cities and municipalities in the Amazon, making several stops at ports and terminals, it is necessary to monitor the route to be taken by the boat, recording the weather conditions and the conditions of the waterways, due to the high volume of production inputs transported on the waterways, which make up the production chain of the Manaus Industrial Estate (PIM) [19,20]. Thus, setting up a data communication infrastructure along the port and river terminals of cities, communities, villages, and towns far from metropolises is very costly [21,22,23].

The motivation behind this work is to provide alternatives for monitoring boats transporting production inputs via long-range communication (LoRa) technology. In this way, the boats carry the data when there is no connection between the ships and transmit the data when they make contact with another boat [24,25,26,27] or with a fixed infrastructure on the riverbed and in the ports or river terminals of the cities [25,28,29,30].

Creating new monitoring and tracking services for boats depends on solving various technical challenges, such as the high mobility of the nodes (boats), which causes variations in link quality and intermittent connectivity [31,32]. In this work, we developed a new hardware platform composed of a system-in-package (SiP)—iMCP HTLRBL32L [33] and GPS, which enables the sharing and transfer of information between boats in a network via LoRa technology. The objective is to investigate the operation of the new hardware platform with the LoRa network in the river environment [32]. The main contributions of this work are as follows:Development of a new hardware platform with a LoRa network and two (2) SiPs. One SiP is used for transmission, and the other SiP is used for data reception. In this way, the boat is ready to transmit and receive LoRa data packets simultaneously;Development of a new end-node-to-end-node communication protocol called the LoRa Protocol Proprietary (LPP), designed to send configurable commands such as a location table, modulation, routing, source DevEUI, destination DevEUI, and port. Additionally, it provides reliability in communication between devices through encryption;Evaluation of the data transfer capacity in a LoRa network in communication scenarios: (i) boat with gateway and (ii) boat with the boat, considering the performance metrics: maximum range, maximum time, received signal strength indication (RSSI), and signal-to-noise ratio (SNR).

The remainder of this paper is organized as follows: Section 2 describes the LoRa/LoRaWAN technology and the SiP iMCP HTLRBL32L. Section 3 discusses related works. Section 4 outlines the steps in building the hardware platform. Section 5 describes the environment where practical experiments were conducted, the configuration parameters of the hardware platform, the river mobility scenario, and the results obtained. Finally, Section 6 presents the conclusions and future directions.

## 2. Related Works

Anwar et al. [34] proposed an algorithm for resource management between end devices and from the device to a LoRaWAN Gateway, considering packet transmission information. Simulation results show the better performance of the proposed RM-ADR algorithm over the G-ADR and ADR algorithms due to faster convergence time, reduced packet loss rate, and retransmission in a mobile LoRaWAN network environment.

Mojamed [35] Investigated the influence of mobility on the performance of a LoRaWAN network using the OMNeT++ simulator. The mobility models used in the simulation were Random Waypoint, Gauss–Markov, and the stationary model. The results show that with a network of up to 5000 end devices moving at different speeds of up to 25 m/s using multiple gateways, the LoRaWAN network performs well, being able to provide good results even at high speeds and larger payloads. With the Random Waypoint mobility model, the results are better in terms of packet delivery rate but at the cost of higher collisions and greater energy consumption.

Parrino et al. [36] proposed a prototype tracking system for asset transportation using GPS and NB-IoT. Position transmission is carried out via NB-IoT using the MQTT protocol. The position is obtained using GPS and NB-IoT. The GPS localization technique is more precise and consumes more energy, while the NB-IoT technique is less precise but has lower energy consumption. Experimental tests were carried out on the road in order to evaluate the tracking error, which depends on the geographical location of the NB-IoT towers.

Sobhi et al. [37] investigate the limits of reliable communication of the LoRaWAN network in terms of speed, throughput, and energy consumption based on practical experiments carried out and computer simulation (ns-3). The results show that reliable communication is achieved at all spreading factors (SFs) for gateway speeds up to 150 km/h with negligible performance degradation at SFs = 11, 12 at speeds above 100 km/h. For synchronous transmission scenarios, the use of lower SFs provides better throughput. However, for semi-synchronized and non-synchronized transmission, higher SFs guarantee better performance, as the longer visibility time allows for a higher probability of gateway detection.

Kim et al. [38] proposed a structure for generating and managing cloud maps, assigning vehicles only the function of transmitting sensor data. The map area was divided into a grid format, and the positions of the elements were adjusted from a computer vision perspective, which was sensitive to external factors such as light and shadow, plus the morphological characteristics of the road elements. By transmitting images with an average size of 350 KB, the implementation made it possible to upload them to the 5G network, with a position accuracy of less than 0.25 m and RMSE regression of less than 0.25 from more than 12,000 elements extracted in 192 partial maps.

Aernouts [39] evaluated location estimation based on received signal strength in an urban area from LPWAN infrastructure with real data and probabilistic location combinations of time difference of arrival (TDoA) and angle of arrival (AoA). In the direct line-of-sight scenario, the average estimation error is 339 m. In the non-line-of-sight scenario, the average error is 159 m.

Araújo et al. [40] evaluated ecosystem services in grazing activities (sheep and goats) with a LoRa network, GPS, and environmental sensors placed on the animals to accurately monitor the extent, intensity, and frequency of grazing. The results demonstrated the high performance and robustness of the IoT system, with minimal data loss, significant battery efficiency, and high resolution on animal movement patterns.

Various studies have evaluated the data transfer capacity of a LoRa network [41,42,43,44,45] in a LoRa network between end devices [46,47,48] and the communication between end devices and the LoRaWAN gateway [27,49,50,51,52,53]. The metrics analyzed are the RSSI [49,54,55,56], SNR [47,55,56,57,58], energy [59], packet reception rate (PDR) [55,60], reception time between packets (PIR) [58,61], and reachability [58,61], among others. However, few studies have focused on tracking and locating boats transporting production inputs along Amazon rivers.

Santos et al. [32] evaluated the data transfer capacity of an ad hoc network consisting of a LoRa transmitter device mounted on a boat and a LoRa receiver device fixed at a riverside house. The boat moved away from and toward the riverside house at speeds of 10, 20, and 30 km/h. The configuration parameters of the LoRa network were as follows: (a) packet size = 54 bytes; (b) frequency band = 915 MHz; (c) transmission bandwidth = 500 kHz; (d) coding rate = 4/5; and (e) spreading factor = 7 bits/s. The results of practical experiments at speeds of 10, 20, and 30 km/h, in both the moving away and approaching scenarios, demonstrate that data transfer via LoRa technology is feasible, with a communication range of up to 1600 m and a contact time of 6 min.

Ortiz et al. [58] analyzed the performance of LoRa technology operating in a vehicular mobility environment through practical experiments and computer simulation (NS-3) [62]. The transmitter unit was fixed inside the vehicle, and transmission began from a distance of 1000 m. The car moved at a constant speed of 30 km/h or 60 km/h toward the fixed receiver unit. The evaluated metrics were the RSSI level, packet reception rate, and interpacket reception time. The LoRa network configuration parameters were as follows: (a) packet size = 47 bytes; (b) frequency band = 915 MHz; (c) transmission bandwidth = 500 kHz; (d) coding rate = 4/5; and (e) spreading factor = 7–12 bits/s. The results show that the practical and simulated experiments were closely aligned and strongly correlated with the evaluated metrics.

Parriet et al. [63] evaluated the communication link quality of a LoRaWAN network in a sea scenario. The two LoRaWAN gateways (receiving devices) installed on land on the property of the company Agroittica Toscana SRL contain receiving antennas located at a height of 13.2 m above sea level. The LoRaWAN Gateway model was the RAK831 LPWAN—RAKWireless [64], driven by a Raspberry Pi 3 model B. The end node (transmitting device) was placed on board a fixed boat on the high seas, with a pole to support the antenna. The measurements were taken with the transmitter antenna at two different heights: H1 = 3.5 m and H2 = 2.1 m above sea level. The STMicroelectronics B-L072Z-LRWAN1 kit was used [65]. The LoRa network configuration parameters were as follows: (a) packet size = 51 bytes; (b) frequency range = 868 MHz; (c) transmission band = 125 kHz; (d) code rate = 4/5; and (e) spread factor = 7–12 bits/s. A Yagi-Uda directional antenna was used [66] with a gain of 9 dBi. The test results show that efficient data transmission can be achieved over a distance of 8.33 km via the worst-case LoRaWAN network configurations in an industrial aquaculture plant scenario. In Table 1, we summarize the main characteristics of the present paper and the most similar related work according to the following aspects: scenario, vehicle, architecture, chip, technology, and approach.

The chips SX1262 [67] and SX1276 [68] are Semtech modules (shown in Table 2) for long-range wireless communication with low power consumption. The SX1262 chip supports a maximum data transfer rate of 300 kbps, with a range of up to 15 km, a power consumption of 6 mA, a reception sensitivity greater than −148 dBm, a transmission power of up to 22 dBm, and an operating frequency range from 150 to b960 MHz. The SX1276 chip, on the other hand, has a maximum data transfer rate of 100 kbps, a range of less than 10 km, a power consumption of 10 mA, a reception sensitivity greater than −142 dBm, a transmission power of up to 20 dBm, and an operating frequency range from 868 to 915 MHz. Importantly, SX1262 offers a higher data transfer rate than SX1276 and greater distances. However, SX1276 has a better signal-to-noise ratio than SX1262 does.

**Table 1 sensors-25-00496-t001:** Comparative table of related works.

Parameters	Santos et al. [32]	Ortiz et al. [64]	Parri et al. [69]	Proposed Work
Scenery	remote communities	urban area	urban area	urban area and remote communities
Vehicle	Boat	Car	Boat	Boat
Architecture	End Node to End Node	End Node to End Node	End Node to Gateway	End Node to End Node and End Node to Gateway
Chip	SX1276	SX1276	SX1276 and RAK831	SX1262
Technology	TTGO T-BEAM ESP32	LoRa RF 96	B-L072Z-LRWAN1	SiP HTLRBL32L
Approach	Experimentation	Experimentation and simulation	Experimentation	Experimentation

**Table 2 sensors-25-00496-t002:** Comparative: SX1262 versus SX1276.

Description	SX1262	SX1276
Data transfer rate	300 (kbps)	100 (kbps)
Range	15 (km)	10 (km)
Power supply	6 (mA)	10 (mA)
LoRa Tx	−148 (dBm)	−142 (dBm)
LoRa Rx	22 (dBm)	20 (dBm)

The new hardware platform is manufactured with the microcontroller and the SX1262 chip integrated and encapsulated, forming the HTLRBL32L SiP. This SiP reduced the size of the newly designed hardware platform, consequently reducing the number of external components and achieving a higher level of integration and project simplification. This work uses two SiPs in the new hardware platform. One SiP is used for transmission, and the other SiP is used for data reception. Why two SIPs?

This work uses two SiPs in the new hardware platform: one SiP is dedicated to transmission, and the other is dedicated to data reception. The decision to use two SiPs is not arbitrary but rather a strategic choice to optimize communication and avoid interference. By separating the transmission and reception functions, the platform ensures better signal quality and lower latency and allows for more precise control over each stage of the communication process. This approach also facilitates the implementation of redundancy mechanisms and enhances system reliability, ensuring that the data-sending and receiving operations are managed efficiently and without conflicts. The use of two SiPs underscores the system’s efficiency and reliability.

The LoRa Protocol Proprietary (LPP) is a game changer in communication between end devices in IoT networks. It enables direct end-node-to-end-node communication without the need for an intermediate gateway at all times. This protocol is designed to send parameterizable commands such as location tables, modulations, routes, source DevEUIs, destination DevEUIs, and ports, offering unprecedented flexibility in network configuration. The ability to adjust these parameters directly on the devices empowers users to dynamically optimize their network, meeting the specific needs of different applications and scenarios. In addition, LPP maintains the robustness and security of communication through advanced encryption, ensuring that transmitted data remain protected against interception and unauthorized access.

The LPP complements and expands the existing LoRaWAN structure, providing a powerful alternative for situations where point-to-point communication is preferred or needed. This adaptation improves network resilience, reduces latency, and increases energy efficiency by eliminating dependence on central gateways. With LPP, end devices can communicate directly, exchanging critical information quickly and securely, which is particularly beneficial in remote monitoring, industrial automation, and tracking applications. This innovation brings a new level of functionality and versatility to IoT networks, enabling the development of more efficient and integrated solutions.

## 3. LoRa/LoRaWAN and the SiP iMCP HTLRBL32L

LoRa [69,70,71,72] is a wireless transmission technology that uses radio frequency based on the chirp spread spectrum (CSS) developed by Semtech. It offers several advantages in long-distance communication, especially in scenarios that require low energy consumption, reliability, and extended coverage. In the context of the Amazon, where communication infrastructure is limited and environments are challenging, LoRa stands out for its characteristics. The main advantages include:

Extended Range: LoRa enables data transmission over several kilometers, making it ideal for large, isolated areas such as the Amazon region. This long-range communication capability is essential for connecting communities and boats in regions without established communication infrastructure, such as communities, riverside villages, and remote river areas.

Low Power Consumption: LoRa is designed to operate with low power consumption, allowing devices to work for long periods using batteries. In remote areas of the Amazon, where access to power is limited and recharging or replacing batteries can be difficult, this energy efficiency is a crucial advantage.

Interference resilience: LoRa uses CSS (Chirp Spread Spectrum) modulation, which provides robust communication against interference. This increases transmission reliability in environments with other radio signal sources and in conditions that affect wave propagation, such as dense forests and large bodies of water, which are common in the Amazon region.

Low Cost: Compared to other long-distance communication technologies, LoRa is affordable in terms of implementation and operating costs. This makes it a viable choice for large-scale projects, especially in regions with limited financial resources where installing sophisticated infrastructure would be unfeasible.

Ability to Operate in Adverse Environments: LoRa has already demonstrated, through experiments and tests, its effectiveness in mobile and difficult-to-access environments, such as rivers, forests, and even coastal areas. This technology has proven reliable when other solutions struggle to maintain constant, long-range coverage.

These benefits justify the choice of LoRa as a communication technology for the Amazon context. Its ability to establish communication networks without complex infrastructure, its extended range, and its low energy consumption make it the ideal solution for connecting boats and riverside communities, even in the most remote locations in the Amazon. In this way, LoRa contributes to the safety, tracking, and monitoring of boats, offering a viable and efficient solution to the logistical and connectivity challenges in the region.

In Brazil, the operating frequency for LoRa devices is 915 MHz, similar to the Australian standard. The modulation parameters for LoRa are as follows:Bandwidth (BW): The BW specifies the frequency range in which the network operates. The BW is set at 125 kHz, 250 kHz, or 500 kHz.The spreading factor (SF) determines the number of symbols representing a single bit of information, ranging from 6 to 12 in LoRa Modulation and 7 to 12 in the Lorawan Protocol.

A higher SF means that each symbol lasts longer, resulting in a greater range and better signal sensitivity as the signals become more robust against noise and interference.
Coding rate (CR): Error checking or correction is performed on the message. The defined values for CR are 4/5, 4/6, 4/7, or 4/8. Increasing the CR provides more protection and increases the number of bits of air.

LoRaWAN (Long Range Wide Area Network) [73,74,75,76] Utilizes the physical modulation of LoRa technology, encoding and decoding the content of received and sent messages. Figure 1 shows the architecture of a LoRaWAN communication system [42,43], which includes the following:

Devices: These devices in the field are responsible for reading the sensors, preprocessing, and sending the read data to the LoRaWAN gateway. They can also control the actuators using the commands received via LoRaWAN.

Gateways: These devices are responsible for communicating via LoRa radio and the LoRaWAN protocol with the end devices and other servers involved in the LoRaWAN architecture securely via the internet. In other words, gateways are intermediaries between LoRaWAN communication and the internet/cloud.

Network server: This server manages communication between the end devices and the application server over the internet. It manages uplink and downlink application messages, adaptive data rate (ADR) functions, and intermediate messages between the join server and the end device in the air activation (OTAA) mode. The join server associates an end device with a LoRaWAN network.

The application server is responsible for handling, managing, and parsing the messages sent by the end devices in the LoRaWAN network, as well as redirecting the messages to external systems and applications.

The difference between LoRa and LoRaWAN is that LoRa handles the type of modulation and physical layer data used between two LoRa end devices or between an end device and a gateway. LoRaWAN, on the other hand, refers to the network architecture (end device, gateways, and servers) in a specific frame (link layer) and datagram (network layer) format, allowing a LoRaWAN end device to transmit data to a LoRaWAN server securely.

The System-in-Package (SiP) iMCP HTLRBL32L—HT Micron (Figure 2) [33,79,80] with the Breakout Board was supplied by the company Hana Electronics [81]. This SiP is designed for IoT network applications and has a microcontroller with integrated and encapsulated LoRa and Bluetooth (BlueNRG-LP) radios. LoRa networks allow data to be sent over long distances with low power consumption. Bluetooth networks send data over short distances at high data rates. The technical specifications of the SiP iMCP HTLRBL32L are shown in Table 3, and the GPS NEO-6M specifications are shown in Table 4.

## 4. Hardware Platform

The platform combines the iMCP HTLRBL32L SiPs and the GPS module to create a flexible and adaptable traceability network in remote areas. This architecture provides options for exchanging data between boats and fixed points, enabling continuous and effective communication along waterways. In the following, we present an overview of the communication scenarios, describe the requirements of the new hardware platform, the baseboard and its block diagram, and the firmware implementation.

### 4.1. Overview

The hardware is powered by two (2) SiPs, each with a distinct role in our communication scenarios. One SiP will transmit the LoRa signal, whereas the other will handle the reception. The communication scenarios are as follows:Boat A leaves port A and, during the journey, receives its GPS position and stores it. Boat A makes a complete journey without encountering another boat. When it arrives at port B (destination port), it transmits all the data stored during the trip to the port B gateway with LoRaWAN, as shown in Figure 3.Boat A leaves port A and meets boat B during the journey, i.e., the boats are within the data communication coverage range. Boat A transmits its stored data and receives data from boat B (Figure 4). The data transmitted and received during data communication between the devices attached to the boats are the boat’s identification, the positions the boat has passed, and the date and time of the positions along the route. If boat A communicates with a fixed infrastructure on the riverbed before boat B, the data from boat A will be downloaded onto the infrastructure. If boat B communicates with the same fixed infrastructure, boat B’s data will be discarded because there is a redundancy of information.Boat A leaves port A, meets boat B, and exchanges information. Boat A continues its journey and meets boat C. In other words, boat A sends information to boat C—its information—and that of boat B, which it meets on its route. (as illustrated in Figure 5). If boat A arrives first at the fixed infrastructure along the riverbed, boat A downloads the data from boats B and C onto the ChirpStack platform. As the information from boats B or C is up to date, if boats B or C download the data, it will be discarded because the information is redundant.Boat A leaves port A and simultaneously meets boats B and C. Boats A, B, and C exchange information about their routes with each other. Assuming that boat A finds a fixed infrastructure along the riverbed, it downloads the data from the three boats onto the ChirpStack platform, as illustrated in Figure 6.

### 4.2. Requirements

Initially, a survey of functional and nonfunctional requirements was conducted for the new hardware platform. The functional requirements address the needs that must be met and resolved by the hardware platform through functions or services, which are as follows:(a)GPS: The hardware must connect to a GPS to obtain the boat’s location information (latitude, longitude, speed, steering angle). The GPS antenna can be internal or external and must guarantee the maximum possible signal coverage along the rivers of the western Amazon.(b)LoRa Radio: Communication between the boat and the infrastructure fixed on the riverbed (LoRaWAN Gateway) and communication between boats must be carried out via LoRa technology via the HTLRBL32L SiP.(c)Bluetooth Radio: The hardware must provide a local connection (near the boat) via Bluetooth Radio via the HTLRBL32L SiP. This connection configures the system, updates the firmware, and extracts data from nonvolatile memory. The hardware must include a connector for the external antenna of the Bluetooth radio.(d)Nonvolatile memory: The hardware must have nonvolatile memory to enable the boat route to be recorded. The recording interval for location points must be configurable.(e)Power Supply: The hardware must receive direct current (DC) power from the boat, from which other voltages required for the various functional blocks will be generated via voltage regulators.(f)Microcontroller: The HTLRBL32L SiP will be used, with two (2) processing units on the board.(g)Communication Interfaces: The hardware must provide communication interfaces compatible with the peripherals: (i) GPS and (ii) nonvolatile memory (SPI or I2C).(h)Connector for writing and debugging: the hardware should provide a connector for writing and debugging the microcontroller(s) firmware.(i)Human Machine Interfaces (HMI): The hardware must have a human–machine interface with the following elements for visual indication: (i) LED for power indication, (ii) LED for activity indication, (iii) LED for LoRa, Bluetooth, and GPS connection status indication, and (iv) LED for fault indication.(j)Watch-Dog-Timer (WDT) and automatic board reset: the hardware must have an automatic reset of the entire system in the event of a persistent fault to recover without the need for operator intervention, keeping its functionalities available.(k)Real-time clock (RTC): The hardware must have a mechanism capable of keeping the system’s date and time updated at all times, even in the absence of power.(l)Power backup: The hardware must provide a connector for the secondary power supply via an external battery to maintain a stable power supply in case of a failure in the boat’s primary power supply. The hardware must include a backup battery charging circuit. The nonfunctional requirements address the characteristics or qualities of the system and are as follows:
(a)Connector/Test points: The board provides test points for the main signals and interfaces in its layout.(b)Usability: The hardware should, whenever possible, be easy to operate and install.(c)Reliability: The hardware should be capable of operating indefinitely and maintaining its functionalities intact, provided that it is kept within the specified operating conditions.(d)Performance: The hardware must be able to handle the peripherals on the card and respond to requests from the interfaces provided, regardless of the order in which they occur, sending a response back to the user.(e)Operating range: A survey of the temperature and humidity ranges along the rivers of the Amazon should be carried out to establish the operating limits of the hardware.(f)Encapsulation: The mechanical encapsulation material and shape must be designed so that the hardware performance is not affected.

### 4.3. Base Plate

The circuits implemented on the board (Figure 7) were developed on the basis of the desired functionalities:

Connectors to the breakout board: Four header female connectors were used on the baseboard to fit the breakout board;

Recording connector: The SiP internal microcontroller is the BlueNRG-LP, and its recording can be performed via the bootloader via an ST-LINK V2-type recorder, which makes it possible to speed up the recording process and use the debugging interface in the Wise Studio software V1.02.02 B02 environment [82], making it easier to develop and test firmware;

GPS module connector: It was necessary to include a connector with a serial communication port for connection to an external GPS. We chose the NEO-6 M GPS module because of its low cost, small size, operation in the same power range as the SiP, and use of the NMEA protocol to exchange messages with the microcontroller;

USB Serial Converter Connector: The SiP has two serial ports. One was dedicated to the GPS, and the other allowed the baseplate to be connected to a computer via a USB serial converter;

Connector with the SPI interface for the MicroSD card module: latitude, longitude, speed, date, and time information had to be stored on and read from the MicroSD card.

Expansion connector with an I2C interface: an optional connector has been inserted on the base plate to obtain information from sensors and actuators;

One power LED and two user LEDs: light indicators (LEDs) have been inserted into the base plate, with the aim of the user identifying whether the plate is on/off with the power LED, as well as checking situations such as the GPS signal, access to the memory card, and use of two general-purpose LEDs;

One push button was used for resetting, and two push buttons were used for general use. As this is a microcontroller system, the first button was provided to reset the system, and the other buttons were used for general use.

### 4.4. Block Diagram

The communication between the boat, the LoRaWAN gateway, and the ChirpStack server is shown in Figure 8, which is as follows:
Boat: The hardware platform is responsible for sending and receiving location information (latitude, longitude, speed, angle, date, and time), both from the boat where the hardware platform is installed and from other boats nearby, and then sending it to the LoRaWAN gateway;The LoRaWAN gateway is responsible for forwarding data packets between the Boats and the cloud computing service. The messages from the boats are sent to the LoRaWAN gateway, which checks the connection to the cloud and sends the data to persist in ChirpStack;ChirpStack server: The LoRaWAN server is responsible for data persistence. When a packet arrives at ChirpStack, an event is triggered, sending the data for persistence in the MariaDB database. The data are stored in this database for use in the web application.

### 4.5. LPP Protocol

The LoRa Protocol Proprietary (LPP) is a new communication protocol between LoRa devices, without packet delivery control, and aimed at communication between boats on Amazonian rivers. The absence of packet delivery control is beneficial for guaranteeing continuous and efficient transmission in difficult-to-access areas, where latency and the absolute reliability of each packet are not the highest priorities. This protocol allows messages to be sent continuously, facilitating communication between boats without requiring a sophisticated network or satellite infrastructure. With LPP, boats can communicate directly and effectively, sending position data, navigation conditions, and other essential information without having to deal with complex retransmissions. This can also be useful in intermittent communication situations, where quick reconnection is more important than resending packets.

The LPP and UDP (User Datagram Protocol) protocols share some similarities when it comes to packet delivery control, but they also have distinct characteristics and uses, which we will highlight:(a)Delivery Control and Reliability—LPP has no delivery control, making it a simple and lightweight solution for communications where guaranteed delivery of each packet is not a priority. It is ideal for communication between boats in remote areas, as it does not waste resources on resending lost packets, keeping the connection as constant as possible.

UDP, on the other hand, does not offer reliability in the delivery of packets, nor does it guarantee their order. However, because it is widely used on IP networks, UDP has its own features and optimizations for the Internet and conventional networks.
(b)Application Environment—LPP is optimized for low-bandwidth networks, such as those found in remote environments. It is especially useful in remote sensing contexts, where constant communication is more important than the exact delivery of each packet. UDP, on the other hand, is geared towards IP networks, such as the Internet, and is used in a wide variety of applications, from gaming and video streaming to IoT sensor communication on Wi-Fi or Ethernet networks. However, it is not suitable for LoRa networks because it does not take into account their bandwidth and range limitations.(c)Power Consumption and Bandwidth—LPP consumes little power, making it more efficient for devices with long-lasting batteries. Thus, LPP maximizes range, even with low bandwidth. While UDP is lighter than TCP, it is not as optimized for low-power devices. It assumes the use of networks with more bandwidth, so it is not ideal for LoRa networks, where power consumption and data transmission capacity are more limited.(d)Development and Customization—LPP is adapted to rustic and specific conditions, such as remote river environments, where prioritizing the simplicity and durability of the system is more important than flexibility of use in different networks. UDP, on the other hand, allows easy integration into various applications and can be combined with other protocols but needs additional customization for efficient use in LoRa networks.

The LPP protocol was designed with 2 SiPs—SiP(Tx) and SiP(Rx)—to support low data rate communications, which is ideal for applications in remote areas where communication infrastructure is limited or non-existent. In the LPP protocol, a state machine was created in the firmware implementation so that the SiP(Tx) is always searching for a LoRaWAN network and, as soon as it finds the network, communicates via IPCP to the SiP(Rx), thus causing the network to switch from the LPP protocol to LoRaWAN. This state remains until the SiP(Rx) sends its entire location table or loses connectivity with the LoRaWAN network.

LPP can use the same nominal frequency band as the LoRa modulation. However, it is advisable to use a frequency slightly different from the channels used for LoRaWAN. For example, in the South American region, the nominal frequency is 915 MHz, with a frequency range ranging from 902.3 MHz to 927.5 MHz. If the channels are in the upper spectrum of the range, it is advisable to use a lower frequency range for LPP, as this will avoid overloading the channels already used by LoRaWAN. In addition to the modulation parameters, LPP also defines a standard frame for its messages, which is contained within the payload field of a standard LoRa frame and has the following fields:-MHDR: Field containing a header indicating the type of LPP message. Currently, it only has the location sending message type, but it can be expanded to up to 255 different message types. FHDR: Field containing the source and destination addresses of an LPP message, which can be used when it is necessary to send unicast messages, that is, from a source to a single destination, with the destination address set to broadcast by default;-Fport: Field containing the port information, together with the MHDR, which can be an option to scale more types of messages;-LPP Payload: Field containing the useful data transmitted in an LPP message; for example, in the Location Send message, this field will contain the Latitude, Longitude, Speed, Date, and Time information;-MIC: Field containing the integrity verification mechanism for LPP messages. This field is optional since a CRC field is already contained in the LoRa modulation, so the MIC would serve as a second integrity mechanism.

The LoRa Proprietary Protocol (LPP) was designed to address specific challenges common in remote and complex environments such as the Amazon. Below are some main issues that conventional communication protocols can face in this region and how LPP can help solve them.
(a)Signal Interference—The Amazon is known for its dense vegetation cover, uneven terrain, and variable weather conditions, which can cause signal interference. Conventional protocols such as Wi-Fi and 4G have difficulty maintaining signal integrity over long distances and in environments with physical obstructions. LPP solves this with a robust modulation of LoRa technology, as it uses a modulation technique called chirp spread spectrum (CSS), which is highly resistant to interference and allows data transmission at low power frequencies. This is ideal for crossing physical obstacles like trees and hills without losing signal quality. Furthermore, this technology has a long range and low frequency, making it less susceptible to environmental interference and able to penetrate dense vegetation better, providing a much greater range than high-frequency protocols.(b)Latency and Communication Delays—In remote regions such as the Amazon, where connectivity to the communication infrastructure is limited, communication delays are frequent, especially in satellite-based networks. For applications involving real-time monitoring, such as boat tracking and environmental monitoring, this latency can compromise the effectiveness of the data. LPP solves this problem by transmitting data directly between devices (point-to-point), eliminating the need for a centralized infrastructure. This drastically reduces latency, as data do not need to pass through multiple network layers before being received. In addition, LPP automatically switches to LoRaWAN mode when it detects an available infrastructure. This allows the device to use long-range networks without constantly relying on them, reducing the need for retransmissions and improving efficiency.(c)Energy Consumption and Sustainability—Most conventional communication protocols require high energy consumption to operate over long distances or in hard-to-reach environments. Energy consumption is a significant concern in Amazon, which has a limited infrastructure. LPP was designed to be highly energy efficient. It transmits data only when necessary, with configurable and random intervals, saving energy and extending the battery life of devices. Thus, instead of maintaining a constant transmission, LPP uses random transmission intervals and a mechanism for periodically checking the LoRaWAN infrastructure. This reduces energy use compared to protocols that always maintain an active connection.(d)Limited Connectivity to LoRaWAN Infrastructure—LoRaWAN gateways can be sporadic and distant in the Amazon. Protocols that rely exclusively on this infrastructure may fail to maintain communication in areas beyond the reach of the gateways. LPP can operate autonomously between devices in point-to-point mode (LPP Mode), i.e., without needing a LoRaWAN infrastructure. This allows devices to communicate directly with each other, sending position updates or collected data, even in areas without network coverage. When a LoRaWAN gateway is detected, LPP automatically switches to this mode, allowing the data stored on the SD card to be uploaded to the central infrastructure. This automatic reconnection system is effective in locations where network coverage is intermittent.(e)Data Reliability and Robustness in Harsh Environments—Conventional protocols may not have security mechanisms to ensure that transmitted data are received and processed. Regarding environmental monitoring and boat tracking, data integrity and reliability are crucial. LPP requires an acknowledgment for each transmitted packet, ensuring that the network server has received the data before proceeding with the transmission of the next packet. This minimizes data loss and increases the reliability of communications. Data can be temporarily stored on an SD card inside the device, ensuring that critical information is not lost even if communication with the infrastructure temporarily fails. When the connection to the LoRaWAN network is re-established, the data are sent to the infrastructure.

### 4.6. Firmware

The circuits implemented on the board (Figure 7) were developed based on the desired functionalities, which are as follows:

The firmware implements the LoRa Proprietary Protocol (LPP) for boat communication. Notably, the transmitter and receiver modules are seamlessly integrated on the same board; i.e., one board has two SiPs. From the transmitting SiP (Tx) perspective, the architecture works according to Figure 9.
Step 1: The Tx transmits the position of the boat obtained by the GPS via the LPP protocol periodically, and the period is generated randomly (interval of 5 to 12 s, with increments every 0.5 s). At the same time, as transmitting the boat’s position, the Tx checks when to switch operating modes, which are LPP or LoRaWAN. LoRaWAN mode identifies a LoRaWAN gateway within the range of Tx. Importantly, the search period for a LoRaWAN infrastructure is configurable and can be up to 300 s (5 min). The default value is 60 s (1 min);Step 2: If there is no response from a LoRaWAN gateway, the Tx returns to LPP operating mode and continues transmitting its position to other boats;Step 3: If the LoRaWAN network is found, Tx starts transmitting the data received from other boats and stored on the SD card;Step 4: If communication with the LoRaWAN network is lost during data transmission, after three attempts, the Tx returns to LPP operating mode and continues transmitting its position until the LoRaWAN network is detected again. For each packet sent to the network server, a confirmation downlink must be received by the Tx to send the next packet;Step 5: If communication with the LoRaWAN network is maintained during the transmission of all the data stored on the SD card, at the end of the transmission, Tx returns to LPP mode, and the process is repeated. Importantly, the SD card data that have been transmitted are deleted from the card once the network server has confirmed that it has been received.

From the point of view of the SiP receiver (Rx), the board works according to the steps described in Figure 10.
Step 1: Rx starts in LPP mode and receives messages from other boats within range;Step 2: If a message is received, the Rx sends this message to the Tx via IPCP (Internet Protocol Control Protocol) communication, which is recorded on the SD card, the file corresponding to the boat that sends the message. The IPCP protocol is responsible for control between the two microcontrollers via the serial port.Step 3: If a problem occurs when the message is received, it is discarded by Rx.

## 5. Experiments

This section covers the environment in which the practical experiments were carried out, the scenarios, the LoRa network configurations, and the results.

### 5.1. Environment and Scenarios

The experiments were carried out at Ceasa port, on the banks of the Negro River, near the Mauazinho, Vila Buriti, and Vila da Felicidade neighborhoods, in the city of Manaus/AM, Brazil. This port is where horticultural products, regional fish, and electrical and electronic products produced in the PIM are shipped to meet the demands of the national and international markets. The connection between the city of Manaus, through the Ceasa port, and the country’s other regions is via ferries and cargo boats that take advantage of the rivers’ course and the region’s geography to ferry vehicles and trucks to the BR 319 highway. Around the port, there is the presence of the Brazilian Navy (Riverine Operations Marine Corps), the Federal Highway Police (Polícia Rodoviária Federal—PRF—in the Portuguese acronym), the Manaus Free Trade Zone Superintendence (SUFRAMA—Superintendência da Zona Franca de Manaus—in the Portuguese acronym), the Manaus Free Trade Zone International Warehouse (EIZOF—Entreposto Internacional da Zona Franca de Manaus—in the Portuguese acronym), and private companies. The practical experiments began at the Ceasa port on the Negro River, crossed the waters of the Negro and Solimões rivers, and went on to the Terra Nova community on the right bank of the Solimões River in Amazonas. This community is located in a floodplain environment, where changes in the landscape due to the seasonal flooding and drought of the rivers directly influence land use and the spatial organization of the economic activities carried out by its residents. The boats used in the experiments are small (speedboats). The practical experimental scenarios (Figure 11) carried out with the LoRa network are as follows:Communication between a boat and a fixed infrastructure on land;Communication between end devices.

The communication range of a LoRa and LoRaWAN network is influenced by environmental and geographic factors that can affect signal propagation. The main factors and possible measures to improve the range and quality of communication are described below:(a)Antenna type: The type of antenna influences electromagnetic radiation levels, depending on the emission power and gain of the antennas. The aim is to maximize the signal performance of the digital antenna, which is suitable for receiving signals from the LoRa network.(b)Antenna location: Antenna positioning is essential for maximizing radio transmission coverage, obtaining the maximum range, and avoiding interference from obstacles. The recommendation is to position the antennas in higher places, such as buildings, towers, and trees, to prevent barriers ahead.(c)Interference in signal transmission and reception: Physical obstacles such as dense forests and houses can obstruct the direct line of sight between the transmitting and receiving devices, limiting communication capacity. As a wireless technology, LoRaWAN is subject to radio frequency interference, which can affect the quality and stability of the data communication link, i.e., network performance. To improve the signal, it is recommended that the gateways be installed in elevated locations (tops of buildings or towers) to reduce the number of obstacles between the transmitter and the receiver. In areas with many obstacles, increasing the number of gateways to cover blind spots and improve the network’s overall coverage is essential. In addition, in the point-to-point communication scenario, directional antennas can focus the signal in a specific direction, reducing interference and increasing the range.(d)Weather conditions: The ambient temperature was 35 °C, the relative humidity was 76%, the wind speed ranged from 0 to 5 km/h, the current speed of the Negro River was 2 km/h, and that of the Solimões River was 4 km/h.(e)Vegetation Density—Dense vegetation (forests, plantations) can absorb and attenuate the LoRa signal, reducing the range. Therefore, placing the antennas above the treetops can prevent the signal from being absorbed by vegetation. A high spreading factor (such as SF 12) is also recommended, which increases the range, although it reduces the data rate.

#### 5.1.1. Communication Between a Boat and a Fixed Infrastructure on Land

The experiments were carried out on the stretch of river delimited by the Negro and Solimões rivers, from the Ceasa port (on the banks of the Negro River) to the Terra Nova community (on the banks of the Solimões River), as shown in Figure 12. A boat with a hardware platform attached moved away from a gateway (s = 0, origin of the spaces) fixed at the Ceasa port until it left the communication coverage range of the LoRaWAN network. The height of the hardware platform was 1.2 m above the river level.

We fixed the gateway at a height of 5 m at the Ceasa port, and the whole experiment started with the boat 10 m away from the port. The boat moved from the Ceasa port to the Terra Nova community and vice versa. The equipment (illustrated in Figure 13) used in the experiment is the (a) base plate; (b) base plate antennas (915 MHz and 6 dBi transmission and reception); (c) 256 GB memory card; (d) NEO Model GPS; (e) batteries (12 volts—7 Ah); (f) Avell A70 HYB I7BS PC, NVidia RTX 3050; (g) WisGate Gateway LoRaWAN Edge Pro—Model RAK7289CV2 IP67; (h) Gateway Antenna—LoRa FiberGlass (900–930 MHz and 6 dBi—transmit and receive). A baseplate has two SiPs and two antennas. Each SiP has an antenna to transmit or receive the radio frequency signal. Table 5 and Table 6 summarize the main configuration parameters of the LoRaWAN network and the gateway used in the experiment. We use the gateway’s adaptive data rate (ADR) mechanism to optimise the data rates, the time the message remains in the air, and the network’s energy consumption. The ADR controls the transmission parameters: The spreading factor, bandwidth, and power. The end device near the gateway used a lower spreading factor and a higher data transmission rate. In comparison, the end device further away from the gateway used a higher spreading factor and a lower data transfer rate.

The average speed in the distance scenario was 20 km/h, with a communication time of approximately 48 min and 24 s and a distance of 16,262 km. There were four repetitions of the practical data transmission experiment between the end device and the gateway. The reference values that guide the RSSI ratio as a function of the SNR in LoRa devices, considering the radio link quality, are shown in Figure 14 [83].

The RSSI and SNR results concerning distance in the practical experiments of the distance scenario are shown in Figure 15a,b. Comparing Figure 14 with Figure 15a,b, we can say the following:”GOOD” for an SNR ≤ −7 during the initial 5964 km, with a communication time of approximately 17 min and 45 s. The communication link had a stable, high-quality connection;”FAIR” for a −7 < SNR < −15, which corresponds to a distance of 5965 km to 11,480 km (equivalent to 5515 km) and a communication time of approximately 16 min and 25 s, the communication link suffered a loss of signal quality; however, the LoRa network was still able to maintain synchronism and operate stably, with the status of ”FAIR”.”BAD” for SNR ≥ −15, i.e., the last 4781 km and a communication time of approximately 14 min and 14 s, the communication link fluctuates between the “FAIR” and “BAD” statuses, with the “BAD” status predominating. Importantly, the environment was very noisy, which impaired the connection and caused the communication link to drop.

The average speed in the approach scenario was 15 km/h, with a communication time of approximately 51 min and 18 s and a distance of 12,928 km. There were four repetitions of the practical data transmission experiment between the end device and the gateway. The RSSI and SNR results for distance in the practical experiments of the approach scenario are shown in Figure 16a,b. Comparing Figure 14 with Figure 16a,b, we can say the following:

The RSSI values ranged from [−114: −46], and the SNRs ranged from [−23: 11]. Considering that the minimum RSSI value in the practical experiment is −114, the SNR value defines the quality classification of the radio link, which can be expressed as follows:For SNR ≥ −15, i.e., the first 3524 km and a communication time of approximately 13 min and 59 s, the communication link suffered severe oscillations, with a predominance of the “BAD” signal, presenting considerable noise, which impaired the connection and caused drops in the communication link.”FAIR” for a −7 < SNR < −15, which corresponds to a distance of 3525 km to 7035 km (equivalent to 3510 km) and a communication time of approximately 13 min and 56 s, the communication link suffers a loss of signal quality. However, the LoRa network was still able to maintain synchronism and operate stably, with a predominance of the “FAIR” status.”GOOD” for an SNR ≤ −7 during the last 5583 km and a communication time of approximately 23 min and 22 s. The communication link had a stable, high-quality connection for sending and receiving data from the LoRaWAN network. Importantly, the communication link sometimes changes to the “FAIR” status and sometimes to the “BAD” status.

#### 5.1.2. Communication Between End Devices

We performed four repetitions of the communication scenario between end devices, as illustrated in Figure 17. We fixed one hardware platform at a height of 7 m in the Ceasa port about the river level, and the other was attached to the moving boat at a height of 1.2 m in relation to the river level. The boat was moving along the Amazon river in the direction of the Terra Nova community, according to the trajectory illustrated in Figure 18, until it left the Loimões network’s communication coverage range. At time t = 0 s, the boat was stationary (v = 0 m/s) and was 10 m away from the fixed hardware platform. At time t = 1 s, the boat started moving toward the Terra Nova community, away from the device fixed on land. Table 7 summarises the main configuration parameters of the LoRa network formed by the end devices.

In the separation scenario, the communication time between end devices is approximately 56 min and 19 s, with a communication distance of approximately 16.262 km, with the boat moving at an average speed of 16 km/h. The results of the RSSI and SNR relative to the distance in the practical experiments in the separation scenario are presented in Figure 19a,b. A comparison of Figure 14 with Figure 19a,b reveals the following:

The RSSI values ranged from [−112: −54], and the SNR ranged from [−15: 14]. Considering that the minimum RSSI value in the practical experiment is −112, the SNR value defines the radio link quality classification, which can be expressed as follows:”GOOD” for an SNR ≤ −7 during the initial 6098 km, with a communication time of approximately 18 min and 9 s. The communication link was stable and of good quality;”FAIR” for a −7 < SNR < −15, which corresponds to a distance of 6099 km up to 8232 km (equivalent to 2133 km), and a communication time of approximately 6 min and 21 s, the communication link suffers a loss of signal quality. However, the LoRa network was still able to maintain synchronism and function stably, oscillating between the “FAIR” and “GOOD” statuses but predominantly with the “FAIR” status.”POOR” for an SNR ≥ −15, i.e., the last 8029 km and a communication time of approximately 23 min and 54 s, the communication link suffered severe oscillations, oscillating between the ”FAIR” and ”BAD” statuses for the most part, but with the ”BAD” status predominating. Importantly, the environment was very noisy, which impaired the connection and caused the communication link to drop.

In the approach scenario, the communication time between end devices is approximately 46 min and 37 s, and the communication distance is approximately 12,447 km, with an average speed of 16 km/h. The RSSI and SNR results in relation to distance in the practical experiments for the approach scenario are shown in Figure 20a,b. Comparing Figure 14 with Figure 20a,b, we can say the following:

The RSSI values ranged from [−113: −43], and the SNRs ranged from [−19: 7]. Considering that the minimum RSSI value in the practical experiment is −113, the SNR value defines the radio link quality classification, which can be expressed as follows:For an SNR ≥ −15, i.e., the first initial 5086 km and a communication time of approximately 19 min and 03 s, the communication link suffered severe oscillations, with a predominance of the “BAD” signal, presenting considerable noise, which impaired the connection and caused drops in the communication link.”FAIR” for a −7 < SNR < −15, which corresponds to a distance of 5087 km to 11,285 km (equivalent to 6198 km) and a communication time of approximately 23 min and 13 s, the communication link suffers a loss of signal quality. However, the LoRa network was still able to maintain synchronism and operate stably with a predominance of “FAIR” status.”GOOD” for an SNR ≤ −7 during the last 1161 km and a communication time of approximately 4 min and 21 s. The communication link had a stable, high-quality connection for sending and receiving data from the LoRaWAN network.

The communication time and range in the Approach scenario are shorter than those in the Away scenario because, initially, the end devices were outside the coverage area, which did not allow messages to be sent and received via the LoRa network, which we configured with a Class C address and OTAA (Over the Air Activation) activation mode. The boat with the LoRa device approached the other boat fixed in the Ceasa port and entered the data communication coverage area. Thus, the transmission and reception of data on the wireless link was carried out with quality and connectivity between the end devices due to the sensitivity and power characteristics of the signal.

## 6. Conclusions and Future Works

The Amazon region, with its vast territory and dense rainforests, faces unique challenges in terms of land infrastructure. The lack of an adequate road network makes transportation by road an unviable option. Adverse weather conditions, such as heavy rains and frequent flooding, worsen the situation, making the few existing roads impassable for long periods of the year. With this difficulty, river navigation has emerged as the primary way of transporting cargo and passengers in the Amazon. The rivers, which cut through the region in a vast interconnected network, offer a natural and efficient solution for transporting goods and people. The rivers of the western Amazon, the Amazon River, the Solimões River, the Madeira River, the Negro River, and many others serve as proper transportation arteries, allowing the movement of large volumes of cargo at significantly lower costs than would be possible by land and air. The efficiency of river transportation, with its ability to transport large amounts of cargo at once, makes it economically viable and more environmentally sustainable than land transportation, which requires greater consumption of fossil fuels and results in higher carbon emissions.

The development of a shipment tracking and monitoring platform using advanced technologies such as GPS and long-range communication networks such as LoRa can support a supply chain across multiple industries by providing a reliable and cost-effective means of moving raw materials and finished goods across the PIM. The platform thus enables real-time tracking of boats, providing critical data on their itinerary, schedules, travel time, adverse conditions, capacity, stopovers, route scheduling, and estimated location. This real-time tracking feature not only improves operational efficiency but also allows for rapid adjustments in response to changing navigation conditions in the event of emergencies, ensuring the system’s reliability.

The introduction of innovative technological solutions, such as the hardware platform developed, which integrates the HTLRBL32L SiP with GPS and LoRa/LoRaWAN technology, represents a significant advance in the logistics supply chain. This platform allows long-distance communication and real-time data transmission, even in the most remote areas of the Amazon. The ability to use two SiPs, one for transmitting and one for receiving data, maximizes the efficiency and reliability of communication, ensuring that boats can be continuously monitored. LoRa Protocol Proprietary (LPP) development, with parameterizable commands and robust encryption, ensures that data are transmitted securely. Owing to its robust encryption ability, LPP ensures that the transmitted data are secure and cannot be intercepted or tampered with, thereby maintaining the integrity and reliability of the system. This innovation has great appeal in the Amazon environment since the integrity and reliability of transmitted and received data are crucial for effectively managing river logistics. Adopting such technologies can significantly reduce operational costs and improve the safety of operations, transparency, and economic, sustainable, and safe development for Amazon.

The limitations of this work are as follows:Dependence on Environmental Conditions—Communication via LoRa in densely wooded river environments, such as the Amazon, can be severely affected by environmental conditions, including physical obstacles (dense vegetation, mountains) and climatic variables (intense rainfall and high humidity). These conditions can interfere with signal propagation, affecting the range and quality of communication. In areas with a high concentration of trees or other natural barriers, signal attenuation can be significant, reducing the effectiveness of monitoring.Distance and Coverage Limitations—Although LoRa is efficient for long-distance communications, it still has range limitations, especially without a direct line of sight between the devices. In scenarios where the mobile device moves away from the fixed infrastructure, signal loss is inevitable beyond a certain distance. This limits continuous monitoring and can lead to the temporary loss of data until the device moves back closer to a gateway or reception point.Lack of Support for High Data Throughput—The LPP protocol and LoRa technology are ideal for transmitting small data packets but are not suitable for applications that require high data throughput. This represents a limitation for monitoring boats that need to send a large amount of information in real-time, as LoRa would not support this level of transfer efficiently. Thus, the approach is limited to scenarios where only essential data, such as location or small pieces of information, need to be transmitted.Energy Consumption in Prolonged Scenarios—Although LoRa has low energy consumption compared to other technologies, prolonged monitoring in remote areas requires careful energy management. The proposed approach may be limited in terms of operational duration, especially if the device needs to operate for long periods without recharging. In locations where access to recharging power is limited, this limitation could compromise the long-term effectiveness of monitoring.Scalability and Reliability in Complex Scenarios—In situations where several devices communicate simultaneously, the scalability of the LoRa network can be limited, as the LPP protocol was not originally designed to support a high number of simultaneous connections in a restricted area. This can lead to interference between devices, packet loss, and deterioration of network reliability in scenarios with a high concentration of boats.

These limitations highlight areas where the proposed system could be improved, such as the addition of redundancy mechanisms, adjustments to the protocol to support a greater volume of data in critical scenarios, and optimizations to better deal with the specific environmental conditions of Amazon. These adjustments could expand the effectiveness and applicability of the approach, providing a more robust and reliable solution for monitoring river transportation in challenging environments.

## 7. Future Works

In future work, we intend to further strengthen data security outside the LoRa/LoRaWAN network by proposing the integration of a private blockchain into the developed platform. Blockchain is known for its ability to provide immutable and auditable records, ensuring that all transactions and tracking data are stored securely and transparently. Each record on the blockchain can be verified and validated in a decentralized manner, eliminating the risk of fraud and tampering. Thus, integrating blockchain with river tracking systems will provide an additional layer of security and trust, essential for critical applications where data accuracy is vital. For example, blockchain ensures that all parties access accurate and unadulterated information in managing boats transporting high-value cargo or sensitive materials, facilitating audits, and increasing trust in the platform. In addition, we intend to expand the connectivity range by exploring hybrid communication technologies, combining LoRa/LoRaWAN with low-orbit satellites, to extend the network’s reach to even more remote areas of the Amazon.

## Figures and Tables

**Figure 1 sensors-25-00496-f001:**
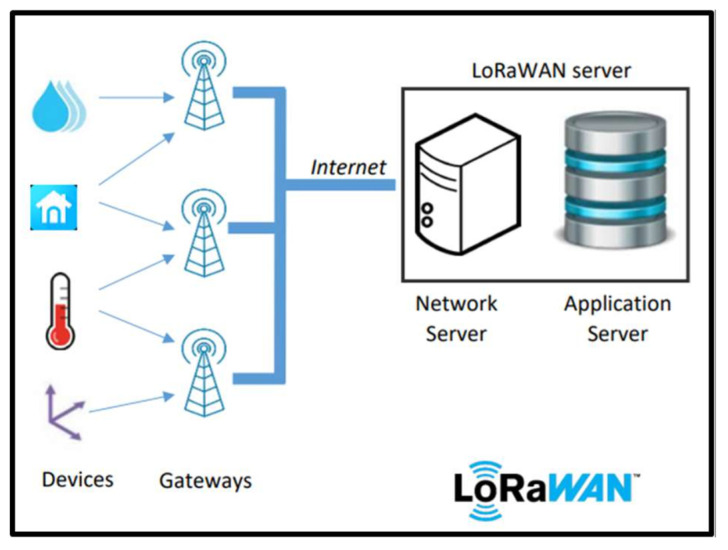
LoRaWAN Communication Architecture [77,78].

**Figure 2 sensors-25-00496-f002:**
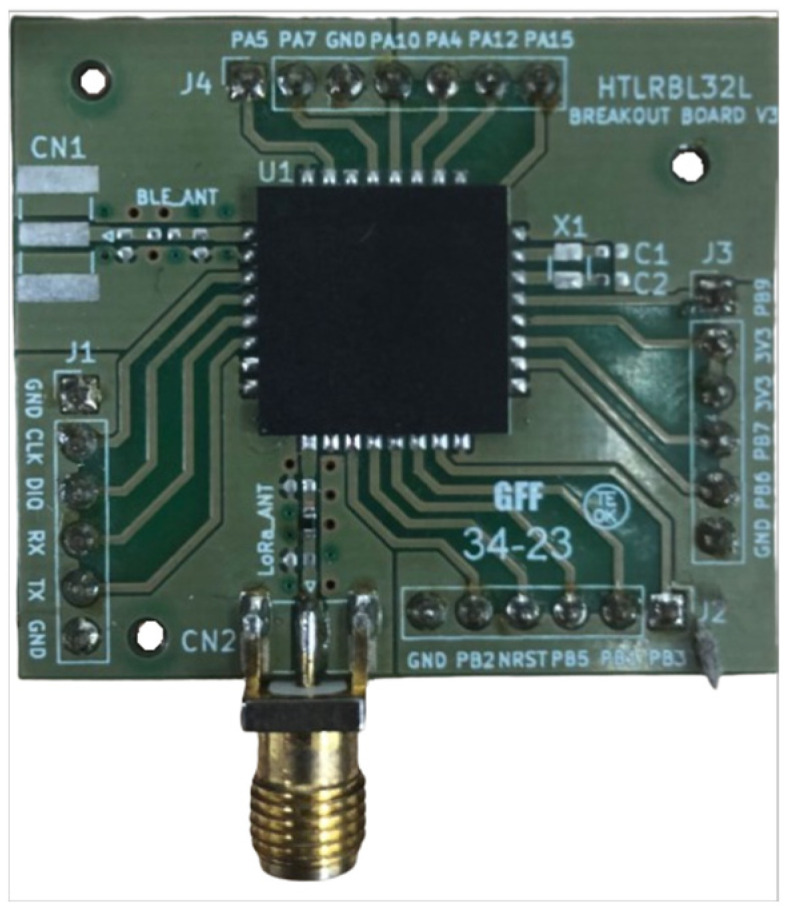
Breakout Board with SiP iMCP HTLRBL32L—HT Micron [33].

**Figure 3 sensors-25-00496-f003:**
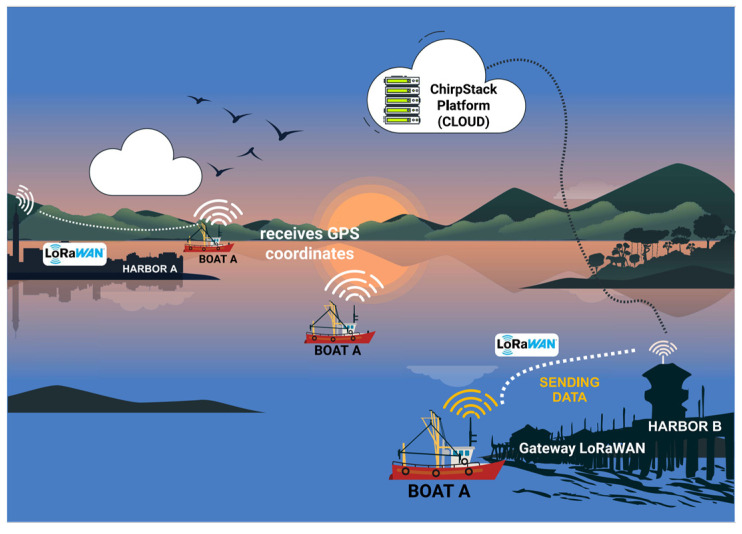
Boat A does not meet any boats on the way.

**Figure 4 sensors-25-00496-f004:**
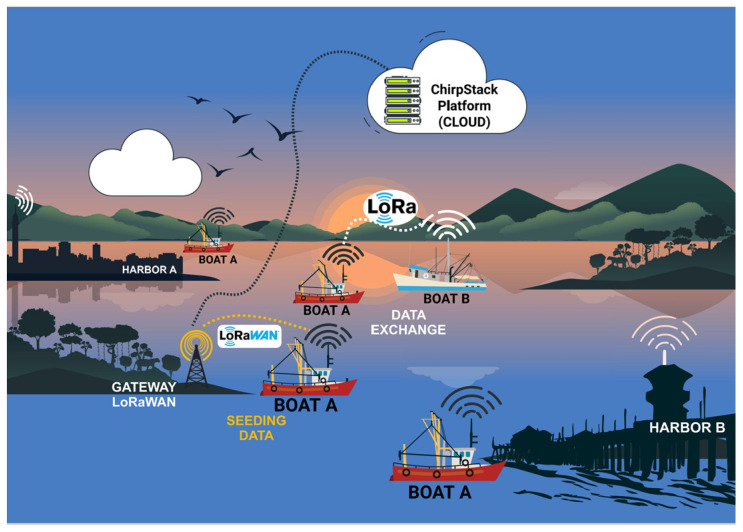
Boat A crosses Boat B during the journey and transmits the data to the fixed infrastructure on the riverbed.

**Figure 5 sensors-25-00496-f005:**
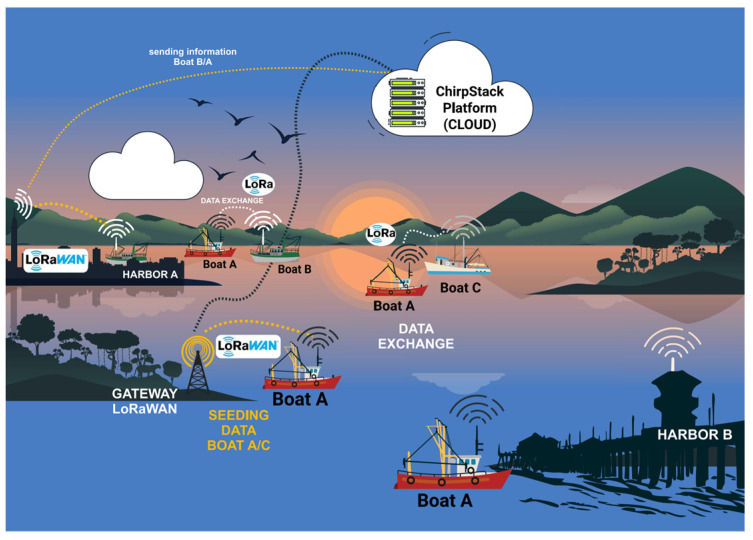
Boat A meets Boat B, and then Boat C.

**Figure 6 sensors-25-00496-f006:**
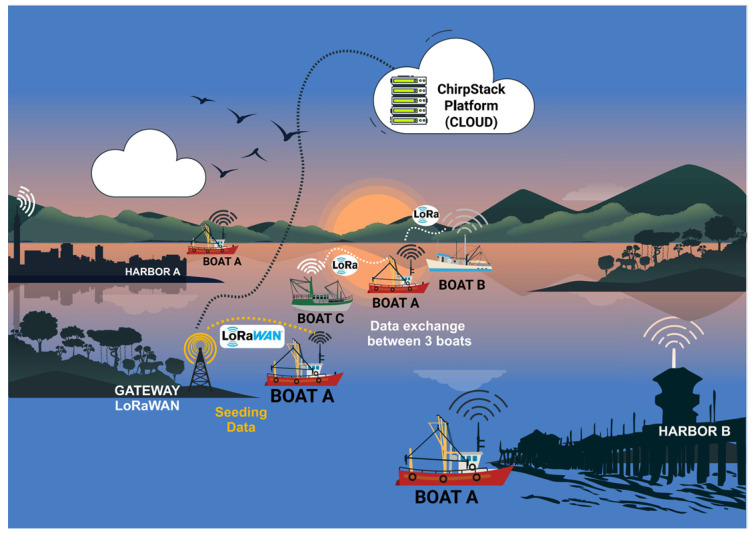
Boat A meets Boats B and C at the same time.

**Figure 7 sensors-25-00496-f007:**
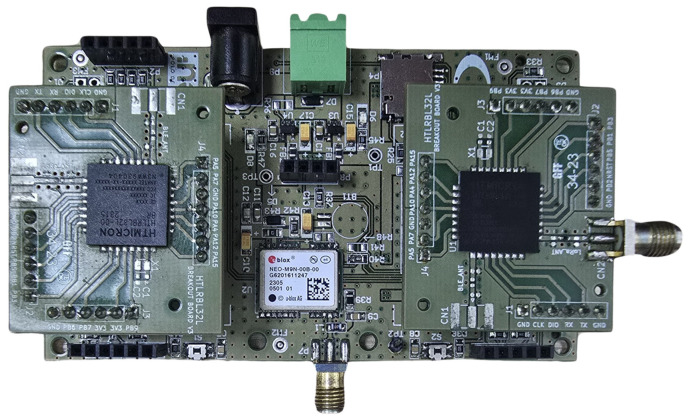
Hardware prototype.

**Figure 8 sensors-25-00496-f008:**
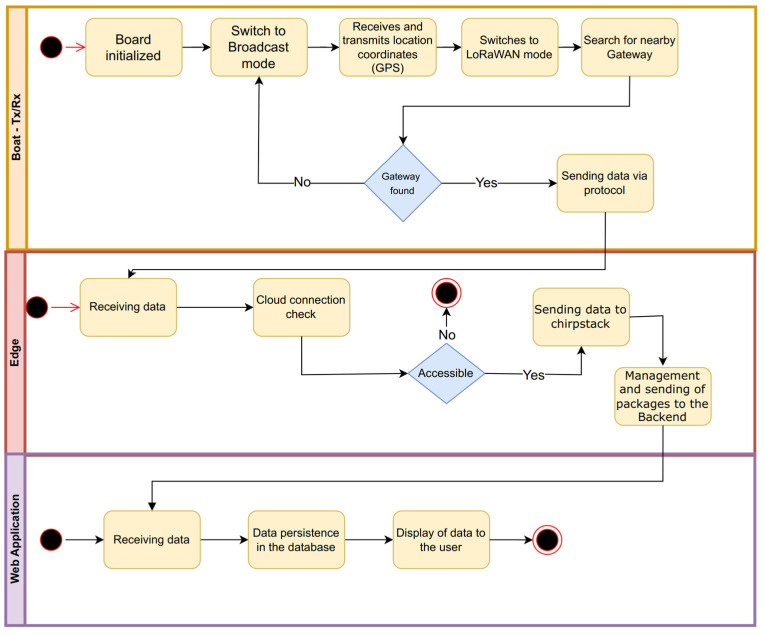
Project Architecture.

**Figure 9 sensors-25-00496-f009:**
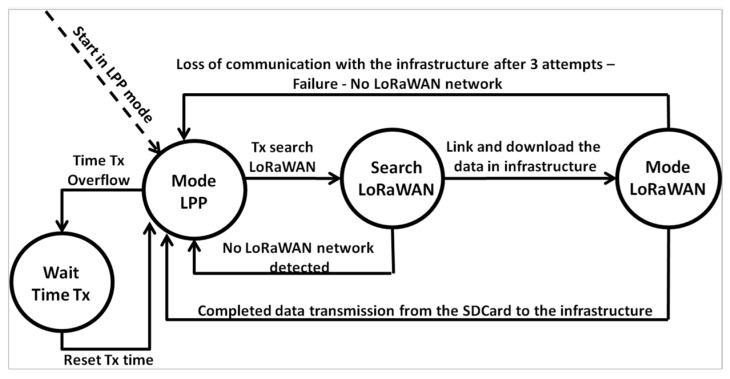
SIP State Machine (Tx).

**Figure 10 sensors-25-00496-f010:**
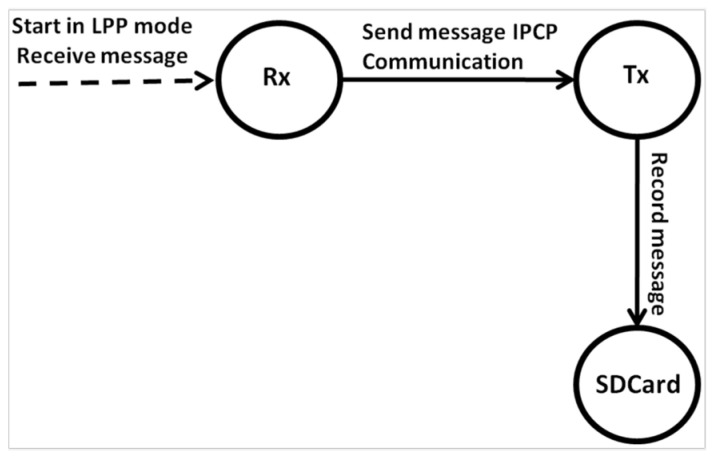
(Rx) SIP State Machine.

**Figure 11 sensors-25-00496-f011:**
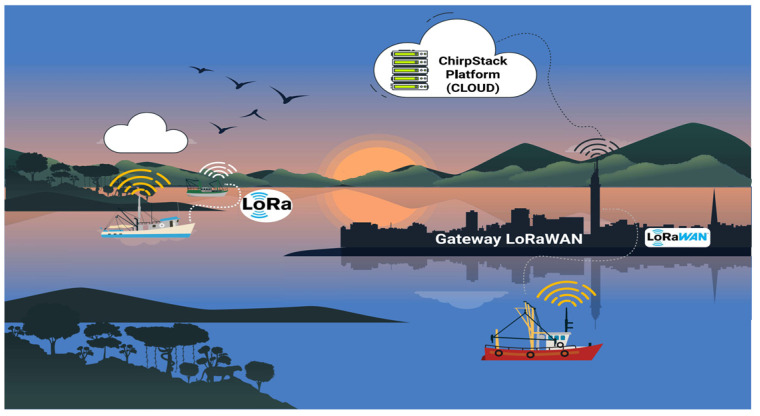
(i) Communication between a boat and a fixed infrastructure on land; (ii) communication between two boats. Practical experiments.

**Figure 12 sensors-25-00496-f012:**
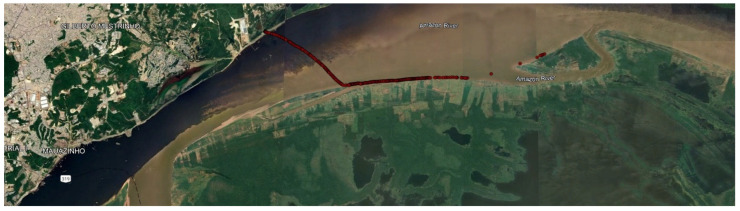
Trajectory between the boat and the gateway.

**Figure 13 sensors-25-00496-f013:**
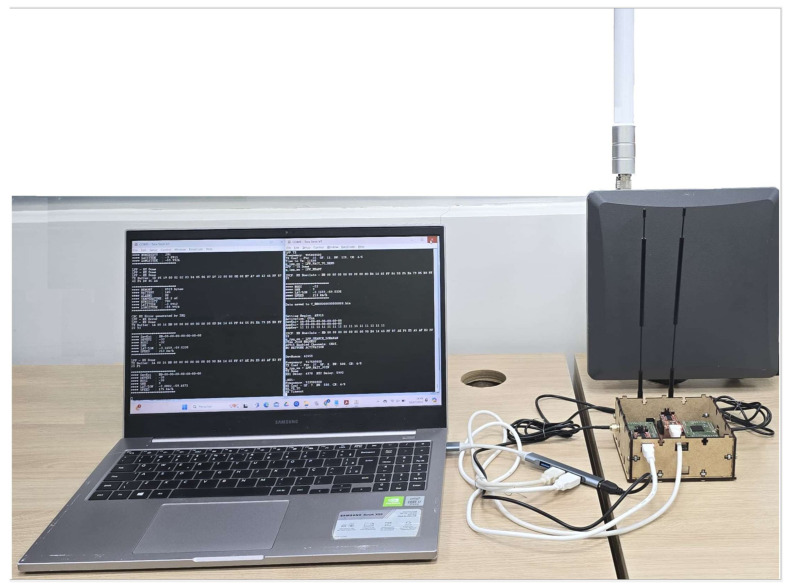
Communication between the hardware platform and the gateway.

**Figure 14 sensors-25-00496-f014:**
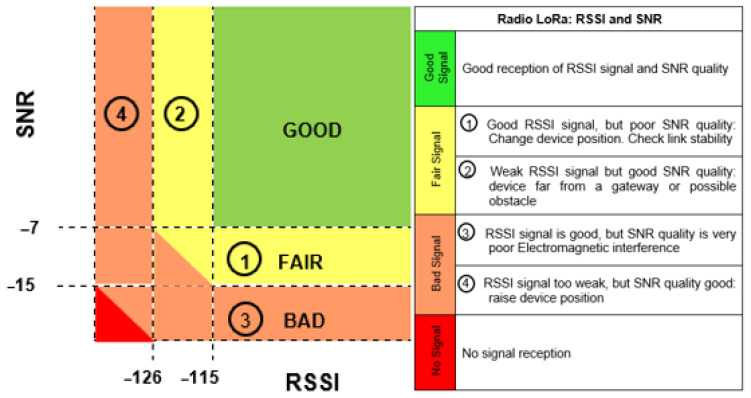
LoRa Network Performance—RSSI X SNR [83].

**Figure 15 sensors-25-00496-f015:**
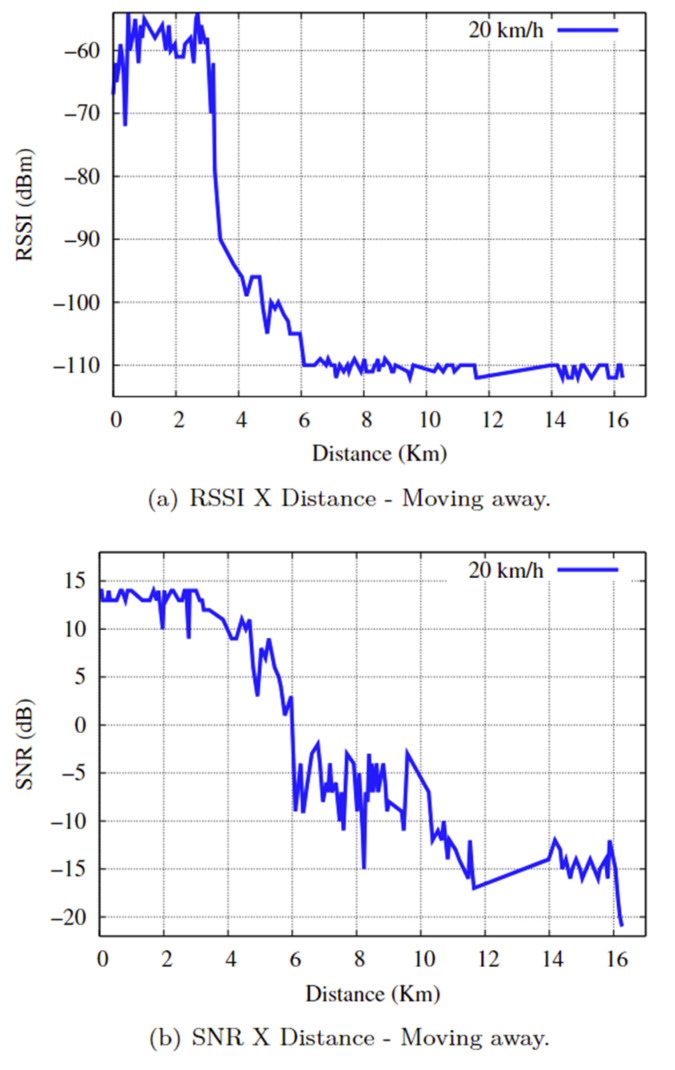
RSSI and SNR as a function of Distance.

**Figure 16 sensors-25-00496-f016:**
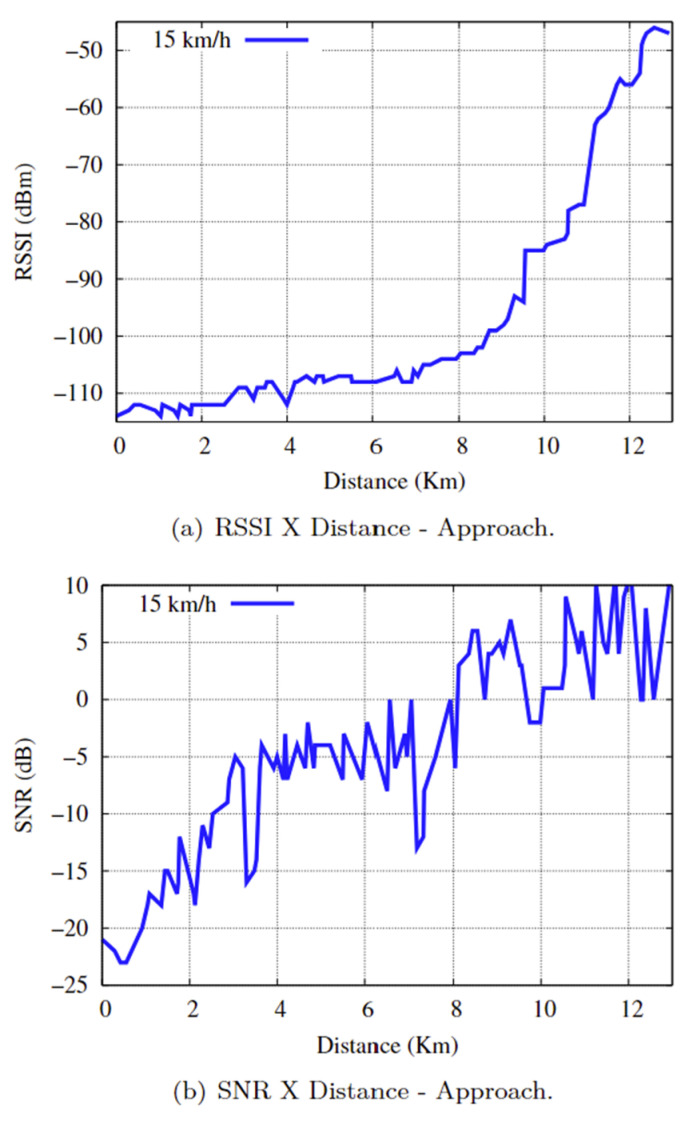
RSSI and SNR as a function of Distance.

**Figure 17 sensors-25-00496-f017:**
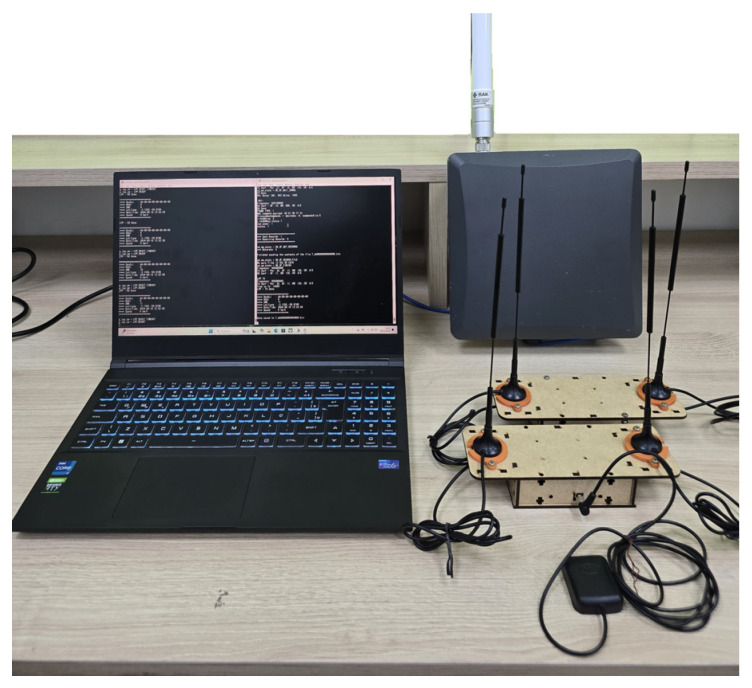
Communication between end devices.

**Figure 18 sensors-25-00496-f018:**
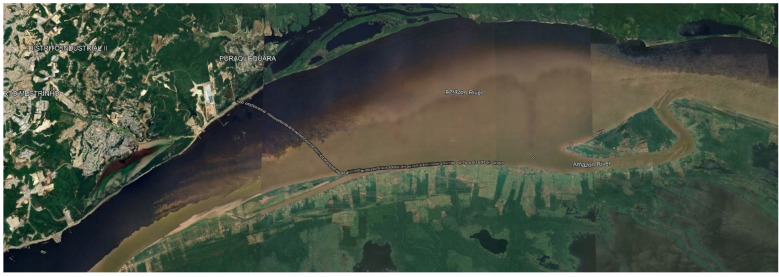
Trajectory between end devices.

**Figure 19 sensors-25-00496-f019:**
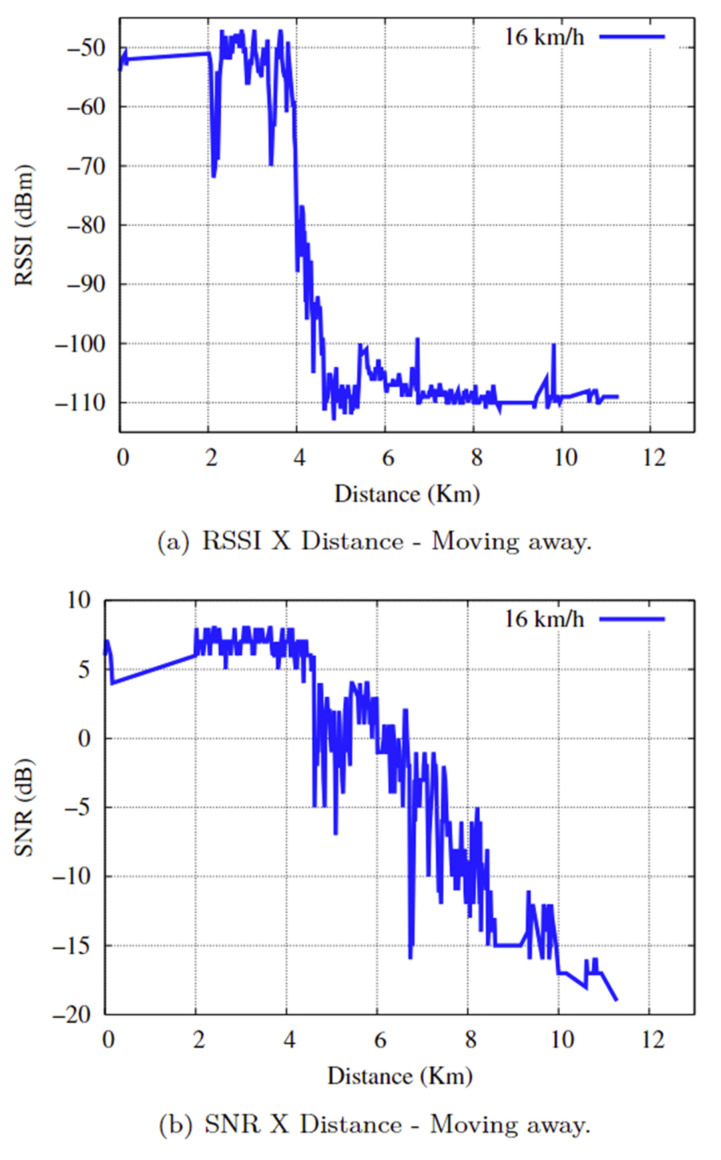
RSSI and SNR as a function of distance.

**Figure 20 sensors-25-00496-f020:**
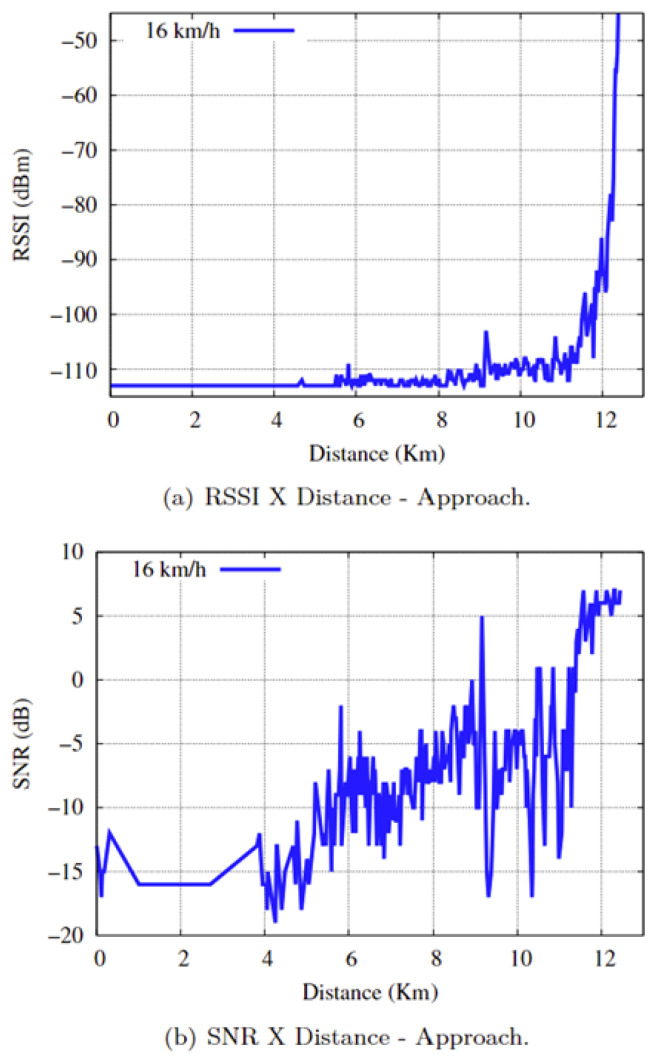
RSSI and SNR as a function of Distance.

**Table 3 sensors-25-00496-t003:** SiP HTLRBL32L features.

Description	Value
MCU and Bluetooth LE Transceiver	STMicroeletronics BlueNRG-355VC
Operation Zones	EU433, EU863-870, US902-928 AU915-928, AS923, KR920-923, IN865-867, RU864-870
Memory	256 KB
Sleep Mode Consumption	2 μA
GPIO	20 pins
Communication Interfaces	USART, LPUART, I2C, SPI, I2S
ADC	5× (12 bits)
Input Voltage	2.7–3.6 V
TX Power	+22 dBm
RX Sensitivity	−132 dBm
Bluetooth TX	+7 dBm
Bluetooth RX	−94 dBm
LoRa TX Consumption	112 mA
LoRa RX Consumption	4.6 mA
Bluetooth TX Consumption	6 mA
Bluetooth RX Consumption	7 mA

**Table 4 sensors-25-00496-t004:** GPS NEO-6M Specifications.

Feature	Specification
Architecture	High-sensitivity GPS receiver
Channels	50 parallel channels
Protocol	NMEA 0183 and UBX
Communication Interface	UART, SPI, USB (some variations)
Horizontal Accuracy	~2.5 m (CEP)
Maximum Speed	500 m/s
Maximum Altitude	50,000 m
Update Rate	Up to 5 Hz (configurable)
Acquisition Time	Cold Start: ~27 s; Hot Start: ~1 s
Operating Voltage	2.7 V to 3.6 V (typical 3.3 V)
Current Consumption	Typical: ~45 mA
Sensitivity (Tracking)	−161 dBm
Sensitivity (Cold Start)	−148 dBm
Sensitivity (Hot Start)	−160 dBm
Dimensions	16 mm × 12.2 mm × 2.4 mm
Weight	~4 g
Operating Temperature	−40 °C to +85 °C
Antenna	Supports active or passive antennas; LNA-integrated
Certifications	CE, FCC, RoHS

**Table 5 sensors-25-00496-t005:** Boat to Gateway–LoRaWAN network.

Description	Value
Package size	[22–242] (bytes)
Frequency	AU 915 (MHz)–8 Channels
Transmission Band	125 (kHz)
Code rate	4/5
Spreading Factor	7–12 (bits/s)
Velocity	15 or 20 (Km/h)

**Table 6 sensors-25-00496-t006:** LoRaWAN Gateway configurations.

Description	Value
Chip	SX 1303 mPCIe card
Channels	8
RX Sensitivity	−139 (dBm)
TX Power	27 (dBm)
Transport Protocol	UDP Packet Forwarder

**Table 7 sensors-25-00496-t007:** Boat-to-boat LoRa network.

Description	Value
Package size	44 (bytes)
Frequency	905 (MHz)
Transmission band	125 (KHz)
Code rate	4/5
Spreading factor	12 (bits/s)
Velocity	16 (km/h)
Protocol	LPP

## Data Availability

Data are contained within the article.

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
