# Peer review of "Tracking Boats on Amazon Rivers—A Case Study with the LoRa/LoRaWAN"

_sensors, 2025, doi:10.3390/s25020496_

Round 1
Reviewer 1 Report
Comments and Suggestions for Authors
See attached file.

Reviewer 2 Report
Comments and Suggestions for Authors
Suggest adding a brief introduction to the current status and challenges of river transportation in the Amazon region at the beginning of the abstract to highlight the necessity and urgency of the research.
Please mention the limitations or deficiencies that may be encountered when applying existing communication technologies in the Amazon region to better illustrate the necessity of developing a new platform.
Technical details:
It is recommended to provide a more detailed description of the specific technical features and roles of SiP iMCP HTLRBL32L and GPS in the platform, so that readers can better understand their innovation and practicality.
Explain the advantages of LoRa technology in long-distance communication and the reasons for choosing LoRa as the communication technology.
Communication protocol:
Provide a more detailed introduction to the LoRa Protocol Proprietary (LPP) protocol, including its design principles, main functions, and advantages.
How can the LPP protocol solve the problems that existing communication protocols may encounter when applied in the Amazon region, such as signal interference, communication delay, etc.
Experimental results:
Provide more details about the experimental environment, methods, and data to enhance the reliability and persuasiveness of the experimental results.
Analyze the experimental results, discuss factors that affect the communication range, such as terrain, weather, etc., and propose possible improvement measures.
Conclusion and Prospect:
Summarize the application prospects and potential value of the new hardware platform and LPP protocol in river transportation safety in the Amazon region.
Propose future research directions, such as further optimizing platform performance and expanding application scope.
Language expression:
Pay attention to the language expression in the abstract to be accurate and concise, and avoid using overly complex or vague vocabulary.
Check for grammar and spelling errors to ensure the accuracy and readability of the abstract.
Round 2
Reviewer 1 Report
Comments and Suggestions for Authors
The paper notably improved after its revision, and I have no further comments. Well done!